# Meeting the Global NCD Target of at Least 10% Relative Reduction in the Harmful Use of Alcohol: Is the WHO European Region on Track?

**DOI:** 10.3390/ijerph17103423

**Published:** 2020-05-14

**Authors:** Charlotte Probst, Jakob Manthey, Maria Neufeld, Jürgen Rehm, João Breda, Ivo Rakovac, Carina Ferreira-Borges

**Affiliations:** 1Heidelberg Institute of Global Health, Universitätsklinikum Heidelberg, 69120 Heidelberg, Germany; 2Institute for Mental Health Policy Research, Centre for Addiction and Mental Health, Toronto, ON M5S 2S1, Canada; neufeldm@who.int (M.N.); jtrehm@gmail.com (J.R.); 3Institute of Clinical Psychology and Psychotherapy, Technische Universität Dresden, 01187 Dresden, Germany; jakobmanthey@snappyquest.org; 4Center for Interdisciplinary Addiction Research, Department of Psychiatry and Psychotherapy, University Medical Center Hamburg-Eppendorf, 20246 Hamburg, Germany; 5Division of Noncommunicable Diseases and Promoting Health through the Life-course WHO European Office for Prevention and Control of Noncommunicable Diseases (NCD Office), 125009 Moscow, Russia; rodriguesdasilvabred@who.int (J.B.); rakovaci@who.int (I.R.); ferreiraborgesc@who.int (C.F.-B.); 6World Health Organization/Pan American Health Organization Collaborating Centre, Centre for Addiction and Mental Health, Toronto, ON M5S 2S1, Canada; 7Institute of Medical Science, University of Toronto, Toronto, ON M5S 1A8, Canada; 8Campbell Family Mental Health Research Institute, Centre for Addiction and Mental Health, Toronto, ON M5G 2C1, Canada; 9Department of Psychiatry, University of Toronto, Toronto, ON M5S 1A8, Canada; 10Dalla Lana School of Public Health, University of Toronto, Toronto, ON M5T 3M7, Canada; 11Department of International Health Projects, Institute for Leadership and Health Management, I.M. Sechenov First Moscow State Medical University, 119992 Moscow, Russia

**Keywords:** alcohol control policy, alcohol consumption, NCD goals, WHO European Region

## Abstract

Background: The Global Action Plan for the Prevention and Control of Noncommunicable Diseases set the target of an “at least 10% relative reduction in the harmful use of alcohol, as appropriate, within the national context”. This study investigated progress in the World Health Organization (WHO) European Region towards this target based on two indicators: (a) alcohol per capita consumption (APC) and (b) the age-standardized prevalence of heavy episodic drinking (HED). Methods: Alcohol exposure data for the years 2010–2017 were based on country-validated data and statistical models. Results: Between 2010 and 2017, the reduction target for APC has been met with a decline by −12.4% (95% confidence interval (CI) −17.2, −7.0%) in the region. This progress differed greatly across the region with no decline for the EU-28 grouping (−2.4%; 95% CI −12.0, 7.8%) but large declines for the Eastern WHO EUR grouping (−26.2%; 95% CI −42.2, −8.1%). Little to no progress was made concerning HED, with an overall change of −1.7% (−13.7% to 10.2%) in the WHO European Region. Conclusions: The findings indicate a divergence in alcohol consumption reduction in Europe, with substantial progress in the Eastern part of the region and very modest or no progress in EU countries.

## 1. Introduction

At 9.8 liter (L) in 2017, the total alcohol per capita consumption (APC) in the World Health Organization (WHO) European Region is the highest of all six WHO regions [1,2]. The region also has the highest prevalence of current drinkers (59.9% of the total adult population) and heavy episodic drinkers (26.4% of the total adult population). On average, drinkers living in the WHO European Region consume 37 g of pure alcohol per day. Finally, with 8.8% of the total adult population, the prevalence of alcohol use disorders is higher than in all other WHO regions [1].

Alcohol use is a causal risk factor for more than 200 health conditions, including many noncommunicable diseases (NCDs), such as neoplasms and cardiovascular diseases [3,4,5]. Alcohol consumption is a known carcinogenic, causally related to cancers in the gastrointestinal tract, liver cancer, and female breast cancer [4,5]. Heavy episodic drinking (HED), in particular, constitutes a major risk factor for cardiovascular outcomes such as ischemic heart disease and ischemic stroke [6,7,8]. Moreover, alcohol consumption has detrimental effects on hypertension, atrial fibrillation, and hemorrhagic stroke, regardless of the drinking pattern [6,7,8].

In the WHO European Region, nearly 90% of mortality in the WHO European Region is from NCDs, and alcohol use makes a sizeable contribution to this mortality burden. For example, in the European Union (EU, plus Norway and Switzerland), a total of 222,426 NCD deaths were caused by alcohol use in 2016, referring to 76% of all alcohol-attributable mortality [9].

With the growing importance of NCDs for the health burden globally, alcohol use has been recognized as a key risk factor. In recent years, alcohol reduction targets have been included in the Sustainable Development Goals (SDG 3.5; 2015–2030) [10], the 13th General Programme of Work of the WHO (2019–2023) [11], and the Global Action Plan for the Prevention and Control of NCDs (2013–2020) [12]. This highlights international recognition of the impact of alcohol on NCD burden, the harm inflicted on individuals and societies, as well as the need to regulate alcohol consumption effectively. SDG target 3.5 is to “strengthen the prevention and treatment of substance abuse, including narcotic drug abuse and harmful use of alcohol”, with APC being a core indicator. The 13th General Programme of Work of the WHO is based on the SDG framework, with a focus on Goal 3, including its targets and indicators regarding alcohol consumption. The Global Action Plan for the Prevention and Control of NCDs sets a clearly defined target of an “at least 10% relative reduction in the harmful use of alcohol, as appropriate, within the national context” by 2025 [12,13]. Progress is to be assessed in the WHO Global Monitoring Framework in comparison to the baseline year of 2010. Three indicators are defined to measure progress: (a) total (recorded and unrecorded) adult APC in liters of pure alcohol consumed per adult 15 years of age and older, (b) the age-standardized prevalence of HED, and (c) alcohol-related morbidity and mortality.

This study aims to explore the progress in achieving the target outlined in the Global Action Plan for the Prevention and Control of NCDs of reducing the harmful use of alcohol by 10% by 2025 concerning two indicators: (a) total adult APC and (b) the age-standardized prevalence of HED by using estimates for the period 2010–2017 at the regional and country levels [13]. Further, this contribution seeks to identify differences in success between groups of countries of the region as well as the challenges and obstacles that impair achievement of the NCD targets.

## 2. Materials and Methods

The first indicator, (a) total APC, was defined as the sum of recorded and unrecorded consumption [14,15], corrected for tourist consumption [16,17]. Unrecorded consumption refers to alcohol that is not accounted for in official statistics on alcohol taxation or sales in the country (such as homemade alcohol, smuggled or illegally produced alcohol, or cosmetic and medicinal products not intended for human consumption [1]). Tourist consumption is estimated as the level of alcohol used by the population going out of the country, adjusted for the consumption of all tourists visiting the given country. The second indicator, (b) HED, was defined here as the age-standardized 30-day prevalence of at least one occasion of drinking 60 g or more pure alcohol.

Changes in APC and the age-standardized prevalence of HED were evaluated among the total adult population as well as among current drinkers. For both indicators, changes were calculated on the regional level for the WHO European Region and country groups, as the relative increase/decrease in percent from 2010 to 2017, by sex. In addition, relative and absolute changes (the absolute increase/decrease in liters of pure alcohol) were evaluated for APC among the total adult population on the country level. Confidence intervals were calculated using bootstrapping accounting for uncertainty in all lower-level estimates. A set of six nondisjoint predefined country groups [18] were used in this analysis. The classification in country groups is based on geography, patterns of substance use, and economic aspects, and is shown in Table 1. Analyses were performed in R version 3.6.1 (R Foundation for Statistical Computing, Vienna, Austria).

Total APC was based on country-validated data for the years 2010 to 2016 as collected for the Global Status Report on Alcohol and Health 2018 [1]. To obtain estimates up to and including 2017, forecasts were obtained from a recent modeling study [16]. In brief, estimates of APC for the presented period are a combination of both collected and forecasted data. Uncertainty intervals around APC estimates take into account the uncertainty linked to the data sources (the more unrecorded alcohol consumption, the higher the uncertainty) and the prediction model (for 2016 and 2017). Prevalence of HED was estimated from survey data, accounting for differences in reporting HED indicators (e.g., 12 months vs. 30 days reference period).

All country-specific data sources and forecast model descriptions in addition to details on calculating uncertainty intervals can be found in the Appendix of Manthey, Rylett, Hasan, Probst, and Rehm [16]. Estimates on alcohol consumption were available for 51 of the 53 member states (no separate estimates were available for the city–state Monaco and the microstate San Marino). HED estimates were age-standardized using population data from the UN Population Division [19] and WHO standard population weights [20]. Regional and country group estimates were calculated as population-weighted averages.

## 3. Results

### 3.1. Total Alcohol Per Capita Consumption

The findings presented in Table 2 show that the target of a −10% reduction in total APC among the total adult population was met by the WHO European Region by 2017. More specifically, APC in the WHO European Region decreased by −12.4% (95% confidence interval (CI) −17.2, −7.0%) from 11.2 L (95% CI 11.0, 11.4) in 2010 to 9.8 L (95% CI 9.3, 10.4) in 2017. This relative decrease was higher among females compared to males. However, it should be noted that in terms of the absolute liters of alcohol consumed, the reduction was higher among males, who reduced their alcohol consumption by about 2 L compared to females with an absolute reduction of 1 L of pure alcohol between 2010 and 2017.

Even though the APC reduction target was overall met, there were considerable differences within the WHO European Region. The largest decreases in APC were observed in the Eastern WHO EUR (−26.2%; 95% CI −42.2, −8.1%) and the South–Eastern WHO EUR (−25.0%; 95% CI −40.3, −7.2%), both of which cut their APC in the total adult population by a quarter. In contrast, no considerable changes in APC among the total adult population were observed in the Central–Western EU (−0.8%; 95% CI −9.0, 7.7%), the Mediterranean (−2.8%; 95% CI −12.8, 7.6%), the EU-28 overall (−2.4%; 95% CI −12.0, 7.8%), and the Central–Eastern EU (−4.5%; 95% CI −18.1, 10.4%).

Figure 1 shows the change in APC in the total adult population between 2010 and 2017 for each country of the WHO European Region. Absolute (in liters of pure alcohol per capita) and relative changes (in percent) are shown, ordered from the highest absolute decrease to the highest absolute increase. The figure shows that a relatively small number of countries were responsible for the overall reduction in the region. With a total adult population of over 140 million, Russia, in particular, made an important contribution towards the regional average with an absolute reduction of about 3 L, referring to a relative reduction by more than −20%. However, the highest absolute decreases were observed in Belarus, Ukraine, and Kyrgyzstan, with a reduction of the APC by more than 5 L. All three countries decreased their APC by more than −35%. On the other end of the spectrum, the three countries with the highest absolute increases by more than 1 L were Latvia, North Macedonia, and Iceland. All three countries increased their APC by more than 10%. Overall, 33 of the 51 countries in the WHO European Region that were included in this study did not meet the target of a 10% relative reduction of the APC.

Among current drinkers, the target of a −10% reduction in APC fell short by 1%. Between 2010 and 2017, a −9.1% (95% CI −14.1, −3.4%) reduction was recorded among current drinkers; however, further reductions in subsequent years may lead to reaching the target by 2025. Overall, the reductions in APC were smaller among current drinkers, with reductions between 0% and −10% in four of the six groups. The strongest reductions in APC among current drinkers were recorded in the South–Eastern WHO EUR (−27.5%; 95% CI −42.3, −10.3%) and the Eastern WHO EUR (−15.8%; 95% CI −34.0, 4.8%).

### 3.2. Heavy Episodic Drinking

Findings regarding the relative change in the age-standardized prevalence of HED, the second indicator for monitoring progress towards the target of reducing the harmful use of alcohol by 10%, are shown in Table 3. Overall, the findings show that little to no progress has been made regarding this indicator, with all CIs including 0% change. The relative change in the prevalence of HED in the adult population of the WHO European Region was −6.4% among females (95% CI −23.7, 10.9%) and −4.8% among males (−13.6, 4.0%). Even in groups with a very high prevalence of HED in the total adult population in 2010, such as the Central–Eastern EU with 44.0% (95% CI 41.2, 46.7%) or the EU-28 with 33.1% (95% CI 30.7, 35.5%), no significant changes were observed.

The largest reduction was recorded for the Eastern WHO EUR, where the age-standardized prevalence of HED in the total adult population declined by −14.3% (95% CI −35.5, 6.9%). A similar picture was observed regarding age-standardized prevalence of HED among current drinkers. In no group did the change of HED among drinkers surpass a −5% reduction, and all CIs included 0%, indicating no significant change between 2010 and 2017. 

## 4. Discussion

Our results show that there is good progress towards achieving the NCD target and that the target of a −10% reduction in the harmful use of alcohol by 2025 has already been achieved in 2017 for total APC in the adult population of the WHO European Region. Consumption among drinkers has also declined in the WHO European Region since 2010, in line with decreases observed in other regions of the world, like the African Region, the Region of the Americas and the WHO Eastern Mediterranean Region [21,22]. Despite reaching the overall target, the WHO European Region still has the highest APC globally, highlighting the need for further decreases. Furthermore, the aggregate picture of the region conceals meaningful differences between countries and country clusters in the region. The observed decline in total APC was strongly driven by strong reductions in member states from the Eastern part of the region, where APC was reduced by about one-quarter. In contrast, no meaningful decline was achieved in the EU-28, the Central–Eastern EU, the Central–Western EU, and countries of the Mediterranean Basin.

However, the findings also show that despite decreases observed globally, no progress has been made in reducing the age-standardized prevalence of HED among the total adult population and current drinkers in the WHO European Region. Of concern, the prevalence of HED remains very high, in particular in parts of the Eastern WHO European Region, and among males. This further underlines the need for evidence-based, coordinated public health efforts and the strengthening of alcohol control policies.

Numerous alcohol control policy options have been shown to reduce alcohol consumption, related health problems and social consequences [22,23,24,25]. The WHO, through the SAFER package, outlines five high-impact strategies that can help governments to reduce the burden of alcohol-related harm [24]: (1) enforcing drunk driving countermeasures; (2) facilitating access to screening, brief interventions, and treatment; (3) restricting the availability of retailed alcohol; (4) raising alcohol prices through excise taxes and pricing policies; and (5) enforcing bans and restrictions on alcohol advertising, sponsorship, and promotion. The latter three interventions are considered to be the most cost-effective and feasible for implementation and are therefore classified as the three “best buys” for preventing and reducing NCD burden stemming from alcohol [22,25].

A broader portfolio of evidence-based policies is included in the WHO European Action Plan for the Prevention and Control of NCDs 2016–2025 in general, and specifically in the European Action Plan to Reduce the Harmful Use of Alcohol 2012–2020. Such alcohol control policy efforts have been intensified in the Eastern part of the WHO European Region and likely contributed to the success in reducing APC. The adoption of core policy measures, most notably the three “best buys” in some countries in the Eastern part of Europe, and their impact, have been well documented, for example, in the Russian Federation [26,27]. In this country, research has shown that alcohol control measures have contributed to an APC reduction of 3.9 L between 2010 and 2016 (in line with the findings shown in this study) [28]. Overall, since the introduction of evidence-based measures, APC could be reduced by 43% between 2003 and 2016, which was linked to an unprecedented rise in Russian life expectancy [28,29]. Moreover, the analysis of a causal link between effective alcohol control policies and mortality has demonstrated that comprehensive government initiatives utilizing evidence-based interventions and intersectoral approaches can notably reduce consumption and related health burdens [30,31]. In Estonia, one of the countries with the highest levels of alcohol consumption in the world 10 years ago, significant progress in reducing alcohol consumption has been achieved after targeting availability, marketing, and taxation of alcohol. Further, the decline in alcohol exposure has been paralleled by sizeable reductions in mortality [32,33]. A continuous decline in APC, alcohol-attributable, and all-cause mortality rates was also observed in Belarus starting from 2011, when a comprehensive national program increased alcohol excise duties, restricted hours of sales, and increased penalties for drunk driving and production of homemade alcohol [34]. Similarly, sustained reductions in alcohol-related traffic harm were observed in Lithuania after implementing a set of evidence-based alcohol control policy measures [35].

The lack of significant progress observed in the EU-28 group is in line with findings reported in the Status Report on Alcohol Consumption, Harm and Policy Responses in 30 European Countries 2019 and the progress evaluation report of the EU Action Plan on Youth Drinking and on Heavy Episodic Drinking [9,36]. According to the latter reports, covering the period 2012–2016, the implementation of the most effective evidence-based policies in the EU member states of the WHO European Region has been hindered by several factors, including limited financial resources, lobbying, and opposition from strong stakeholders, foremost the alcohol industry. Moreover, policy decisions in other areas may have contributed to the lack of progress in reducing alcohol consumption in the EU-28. For example, government subsidies for alcohol production, which can make alcohol less expensive, undercutting the effectiveness of tax systems on alcohol consumption, have increased in several countries. To the extent by which such subsidies support exports, but also increase affordability and marketing of alcoholic beverages, they can also undermine the efforts of other countries to reduce alcohol-related harm [37,38] and lead to changes in patterns of consumption, specifically in young people [39].

While there is strong evidence that alcohol policies impact drinking behavior on the population level, other factors may also have influenced changes—or the lack thereof—in alcohol consumption in the region. Up to a certain country income level, the gross domestic product of a country is strongly associated with the prevalence and level of alcohol consumption [40,41]. Changes in the economy and employment rates may have contributed to changes in alcohol consumption observed in this study [42,43,44,45]. Other factors such as changes in public attitudes and social norms around drinking may also play a role but they are entwined with the effects of alcohol policies and difficult to measure. The present study has a few noteworthy limitations. First of all, alcohol-related morbidity and mortality, the third indicator, was not analyzed separately. The consideration of regional changes in alcohol-attributable harm might have led to different conclusions as the alcohol-attributable burden is overall higher in the Eastern parts of the region. An overview of changes in health burden attributable to substance use in the WHO European Region can be found elsewhere [46]. Second, the study analyzed changes in APC and the prevalence of HED from 2010 to 2017 on a descriptive level. Conclusions regarding the underlying driving factors (such as alcohol policy changes or the lack thereof) are based on reasoning rather than data directly, and more research is needed at the country and regional levels to establish causal links.

## 5. Conclusions

In summary, our findings indicate that there is a divergence in the reduction of alcohol consumption, with the Eastern part of the region—and countries of the former Soviet Union in particular—pulling its weight and the EU-28 making very modest or no progress at all, with very few exceptions [47]. The majority of the countries (33 out of 51) in the WHO European Region that were included in this study did not meet the target of an at least 10% relative reduction in the harmful use of alcohol in either of the two indicators investigated. A swift and purposeful implementation of evidence-based policy changes will be required in these latter countries to reach the target by the year 2025.

## Figures and Tables

**Figure 1 ijerph-17-03423-f001:**
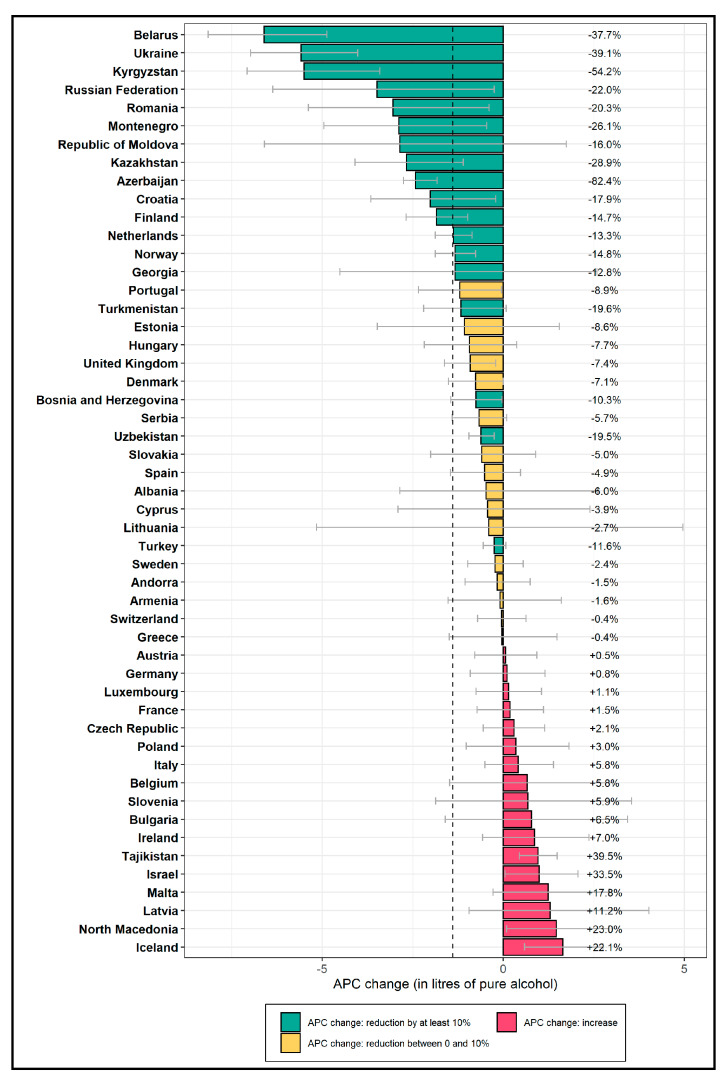
Changes in alcohol per capita consumption (APC) in the total adult population between 2010 and 2017, by country. Absolute changes are shown on the x-axis in liters of pure alcohol; relative changes are indicated as percentages on the right of the graph and in the color-coding. Countries that are shown in green already achieved the target of an at least −10% reduction in APC. The dashed vertical line indicates the mean absolute change in the WHO European Region.

**Table 1 ijerph-17-03423-t001:** Country groups used in the analysis [18].

Country Groups	Countries
EU-28	Austria, Belgium, Bulgaria, Croatia, Cyprus, Czech Republic, Denmark, Estonia, Finland, France, Germany, Greece, Hungary, Ireland, Italy, Latvia, Lithuania, Luxembourg, Malta, Netherlands, Poland, Portugal, Romania, Slovakia, Slovenia, Spain, Sweden, United Kingdom
Central–Eastern EU	Bulgaria, Croatia, Czech Republic, Estonia, Hungary, Latvia, Lithuania, Poland, Romania, Slovakia, Slovenia
Central–Western EU	Austria, Belgium, Denmark, Germany, Netherlands, Switzerland
Eastern WHO EUR	Belarus, Republic of Moldova, Russian Federation, Ukraine
Mediterranean	Cyprus, France, Greece, Israel, Italy, Malta, Portugal, Spain, Turkey
South–Eastern WHO EUR	Armenia, Azerbaijan, Georgia, Kazakhstan, Kyrgyzstan, Tajikistan, Turkey, Turkmenistan, Uzbekistan

Note: EU—European union; WHO—World Health Organization; EUR—European Region.

**Table 2 ijerph-17-03423-t002:** Alcohol per capita consumption (APC) in liters of pure alcohol in 2010 and 2017 and relative change in APC in the World Health Organization European Region and six nondisjoint country groups among the total adult population and current drinkers.

		Population			Drinkers		
Country Group	Sex	APC 2010 (95% CI)	APC 2017 (95% CI)	% Change (95% CI)	APC 2010 (95% CI)	APC 2017 (95% CI)	% Change (95% CI)
WHO European Region	Female	5.1 (5.0, 5.2)	4.3 (4.1, 4.6)	−14.9% (−19.5, −9.6%)	8.9 (8.7, 9.0)	7.9 (7.5, 8.4)	−10.6% (−15.4, −5.0%)
	Male	17.5 (17.2, 17.8)	15.4 (14.5, 16.3)	−12.3% (−17.1, −6.8%)	24.0 (23.6, 24.4)	21.7 (20.5, 23.0)	−9.9% (−14.9, −4.3%)
	Total	11.2 (11.0, 11.4)	9.8 (9.3, 10.4)	−12.4% (−17.2, −7.0%)	17.3 (17.0, 17.6)	15.7 (14.9, 16.7)	−9.1% (−14.1, −3.4%)
Central–Eastern EU	Female	5.9 (5.7, 6.1)	5.6 (4.8, 6.4)	−6.1% (−19.5, 8.6%)	8.7 (8.4, 9.0)	8.1 (7.0, 9.4)	−6.3% (−19.7, 8.3%)
	Male	19.6 (19.0, 20.3)	18.7 (16.0, 21.6)	−5.0% (−18.5, 9.9%)	23.6 (22.8, 24.4)	22.3 (19.2, 25.8)	−5.4% (−18.8, 9.4%)
	Total	12.6 (12.2, 13.0)	12.0 (10.3, 13.9)	−4.5% (−18.1, 10.4%)	16.7 (16.2, 17.2)	15.9 (13.6, 18.4)	−4.8% (−18.4, 10.0%)
Central–Western EU	Female	5.8 (5.7, 6.0)	5.8 (5.3, 6.3)	−1.1% (−9.2, 7.4%)	8.0 (7.8, 8.1)	7.7 (7.1, 8.4)	−2.7% (−10.7, 5.6%)
	Male	19.0 (18.6, 19.4)	18.6 (17.1, 20.2)	−1.9% (−10.0, 6.5%)	21.8 (21.3, 22.3)	21.2 (19.5, 23.0)	−2.7% (−10.7, 5.6%)
	Total	12.3 (12.0, 12.5)	12.2 (11.2, 13.2)	−0.8% (−9.0, 7.7%)	15.3 (15.0, 15.7)	15.0 (13.8, 16.3)	−2.1% (−10.1, 6.3%)
Eastern WHO EUR	Female	6.9 (6.6, 7.2)	4.8 (3.7, 5.9)	−31.3% (−46.1, −14.5%)	12.3 (11.8, 12.8)	10.1 (7.9, 12.6)	−17.9% (−35.6, 2.1%)
	Male	25.7 (24.6, 26.9)	19.4 (15.2, 24.1)	−24.8% (−41.1, −6.3%)	33.9 (32.4, 35.4)	27.9 (21.9, 34.8)	−17.6% (−35.4, 2.6%)
	Total	15.6 (14.9, 16.3)	11.5 (9.0, 14.3)	−26.2% (−42.2, −8.1%)	23.9 (22.9, 25.0)	20.1 (15.8, 25.1)	−15.8% (−34.0, 4.8%)
EU-28	Female	5.4 (5.3, 5.5)	5.2 (4.7, 5.8)	−3.5% (−13.0, 6.5%)	7.8 (7.6, 8.0)	7.5 (6.8, 8.3)	−3.4% (−12.9, 6.7%)
	Male	17.9 (17.4, 18.4)	17.4 (15.7, 19.2)	−2.9% (−12.5, 7.1%)	21.2 (20.6, 21.8)	20.6 (18.5, 22.7)	−3.0% (−12.5, 7.1%)
	Total	11.5 (11.2, 11.8)	11.2 (10.1, 12.4)	−2.4% (−12.0, 7.8%)	15.0 (14.6, 15.4)	14.6 (13.2, 16.2)	−2.4% (−12.0, 7.8%)
Mediterranean	Female	3.6 (3.5, 3.7)	3.5 (3.1, 3.9)	−3.9% (−13.5, 6.1%)	7.1 (6.9, 7.3)	7.1 (6.4, 7.8)	−0.5% (−10.5, 9.9%)
	Male	12.7 (12.4, 13.1)	12.3 (11.1, 13.7)	−3.2% (−13.2, 7.2%)	19.5 (19.0, 20.1)	19.4 (17.4, 21.4)	−0.8% (−11.0, 9.9%)
	Total	8.0 (7.8, 8.3)	7.8 (7.0, 8.6)	−2.8% (−12.8, 7.6%)	13.9 (13.5, 14.2)	13.9 (12.4, 15.3)	0.1% (−10.2, 10.8%)
South–Eastern WHO EUR	Female	1.4 (1.3, 1.5)	1.0 (0.8, 1.3)	−27.6% (−42.9, −9.7%)	11.0 (10.5, 11.5)	7.8 (6.1, 9.7)	−29.1% (−44.1, −11.6%)
	Male	6.7 (6.4, 7.1)	5.1 (4.0, 6.2)	−25.0% (−40.1, −7.5%)	29.3 (27.9, 30.7)	21.1 (16.8, 26.0)	−28.1% (−42.6, −11.2%)
	Total	3.9 (3.8, 4.1)	3.0 (2.4, 3.7)	−25.0% (−40.3, −7.2%)	22.3 (21.2, 23.3)	16.1 (12.8, 20.0)	−27.5% (−42.3, −10.3%)

Note: APC—alcohol per capita consumption; CI—confidence interval; WHO—World Health Organization; EU—European Union (including the United Kingdom); EUR—European Region.

**Table 3 ijerph-17-03423-t003:** Age-standardized prevalence of heavy episodic drinking (HED) in 2010 and 2017 and relative change in the prevalence of HED in the World Health Organization European Region and six nondisjoint country groups among the total adult population and current drinkers.

		Population			Drinkers		
Group	Sex	HED 2010	HED 2017	% Change	HED 2010	HED 2017	% Change
WHO European Region	Female	19.6% (15.8, 23.5%)	18.4% (15.0, 21.7%)	−6.4% (−23.7, 10.9%)	34.2% (27.5, 41.0%)	33.7% (27.5, 39.9%)	−1.6% (−19.8, 16.5%)
	Male	41.8% (38.0, 45.6%)	39.8% (36.1, 43.4%)	−4.8% (−13.6, 4.0%)	57.3% (52.1, 62.6%)	56.1% (50.9, 61.3%)	−2.2% (−11.3, 6.8%)
	Total	30.7% (26.9, 34.5%)	29.1% (25.5, 32.6%)	−5.4% (−16.9, 6.1%)	47.5% (41.6, 53.4%)	46.7% (41.0, 52.3%)	−1.7% (−13.7, 10.2%)
Central–Eastern EU	Female	30.3% (27.5, 33.0%)	31.7% (28.8, 34.5%)	4.6% (−4.6, 13.9%)	44.3% (40.2, 48.3%)	46.2% (42.1, 50.3%)	4.3% (−4.9, 13.6%)
	Male	57.7% (54.9, 60.4%)	58.9% (56.2, 61.6%)	2.0% (−2.6, 6.7%)	69.3% (66.0, 72.6%)	70.5% (67.2, 73.7%)	1.6% (−3.0, 6.3%)
	Total	44.0% (41.2, 46.7%)	45.3% (42.5, 48.0%)	2.9% (−3.3, 9.2%)	58.3% (54.6, 61.9%)	59.8% (56.1, 63.4%)	2.6% (−3.7, 8.8%)
Central–Western EU	Female	19.8% (17.8, 21.8%)	21.0% (18.9, 23.1%)	6.0% (−4.7, 16.7%)	27.0% (24.3, 29.8%)	28.2% (25.3, 31.0%)	4.2% (−6.3, 14.7%)
	Male	45.1% (42.2, 48.0%)	46.3% (43.1, 49.4%)	2.6% (−4.4, 9.6%)	51.8% (48.4, 55.1%)	52.7% (49.1, 56.3%)	1.8% (−5.1, 8.7%)
	Total	32.5% (30.0, 34.9%)	33.6% (31.0, 36.3%)	3.6% (−4.5, 11.7%)	40.6% (37.5, 43.6%)	41.5% (38.3, 44.8%)	2.3% (−5.7, 10.3%)
Eastern WHO EUR	Female	27.9% (17.3, 38.4%)	23.1% (14.2, 32.1%)	−17.0% (−49.2, 15.3%)	49.5% (30.8, 68.2%)	49.1% (30.1, 68.2%)	−0.7% (−39.3, 37.8%)
	Male	53.6% (44.9, 62.3%)	46.7% (38.4, 55.0%)	−12.9% (−28.4, 2.5%)	70.6% (59.2, 82.0%)	67.3% (55.4, 79.3%)	−4.6% (−21.6, 12.3%)
	Total	40.7% (31.1, 50.3%)	34.9% (26.3, 43.5%)	−14.3% (−35.5, 6.9%)	62.4% (47.7, 77.2%)	61.0% (45.9, 76.1%)	−2.3% (−26.5, 21.9%)
EU-28	Female	20.6% (18.5, 22.6%)	20.8% (18.8, 22.9%)	1.3% (−8.6, 11.3%)	29.7% (26.8, 32.7%)	30.2% (27.2, 33.1%)	1.5% (−8.5, 11.4%)
	Male	45.7% (42.9, 48.4%)	45.7% (42.9, 48.5%)	0.0% (−6.0, 6.1%)	54.1% (50.9, 57.3%)	54.1% (50.8, 57.4%)	−0.0% (−6.1, 6.1%)
	Total	33.1% (30.7, 35.5%)	33.3% (30.9, 35.7%)	0.4% (−6.8, 7.7%)	43.3% (40.2, 46.4%)	43.5% (40.3, 46.6%)	0.5% (−6.8, 7.8%)
Mediterranean	Female	12.4% (11.1, 13.7%)	11.8% (10.5, 13.1%)	−4.6% (−14.9, 5.7%)	24.2% (21.7, 26.8%)	23.9% (21.4, 26.5%)	−1.2% (−11.9, 9.4%)
	Male	31.0% (28.9, 33.2%)	29.9% (27.8, 32.1%)	−3.6% (−10.6, 3.3%)	47.5% (44.2, 50.9%)	47.0% (43.6, 50.3%)	−1.2% (−8.3, 5.9%)
	Total	21.7% (20.0, 23.4%)	20.9% (19.2, 22.6%)	−3.8% (−11.7, 4.1%)	37.5% (34.5, 40.5%)	37.1% (34.1, 40.1%)	−1.0% (−9.1, 7.2%)
South–Eastern WHO EUR	Female	3.9% (3.4, 4.3%)	3.9% (3.4, 4.3%)	−0.3% (−12.4, 11.7%)	30.4% (26.9, 33.9%)	29.6% (26.0, 33.2%)	−2.5% (−14.2, 9.3%)
	Male	11.5% (10.5, 12.5%)	11.5% (10.4, 12.5%)	−0.5% (−9.9, 8.9%)	50.1% (45.7, 54.6%)	47.8% (43.3, 52.4%)	−4.6% (−13.6, 4.5%)
	Total	7.7% (7.0, 8.5%)	7.7% (6.9, 8.5%)	−0.2% (−10.2, 9.8%)	43.6% (39.5, 47.8%)	42.1% (37.8, 46.3%)	−3.6% (−13.3, 6.1%)

Note: HED—Heavy episodic drinking; CI—confidence interval; WHO—World Health Organization; EU—European Union (including the United Kingdom); EUR—European Region.

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
