# Peer review of "Meeting the Global NCD Target of at Least 10% Relative Reduction in the Harmful Use of Alcohol: Is the WHO European Region on Track?"

_ijerph, 2020, doi:10.3390/ijerph17103423_

Round 1
Reviewer 1 Report
Alcohol is a psychoactive substance with dependence-producing properties that has been widely used in many cultures for centuries. The harmful use of alcohol causes a large disease, social and economic burden in societies. The health, safety and socioeconomic problems attributable to alcohol can be reduced and requires actions on the levels, patterns and contexts of alcohol consumption and the wider social determinants of health. Countries have a responsibility for formulating, implementing, monitoring and evaluating public policies to reduce the harmful use of alcohol. Therefore, presented manuscript concerns a very current and important problem. The entire manuscript is written in accordance with the requirements of the journal. However, there are several weaknesses and the improvement of them can significantly increase value of manuscript.
- In introduction the Authors claimed that “adult per capita consumption in liters of pure alcohol consumed per adult 15 years of age and older” but 15-year-old is not an adult. This sentence should be corrected.
- If the authors use an abbreviation once, they should use it everywhere.
- Please explain the purpose of description from line 60 to line 70. In my opinion it should be deleted.
- Introduction is too short and general, it contains too few messages. It should be extended and include more information’s.
- Lines 60 to 80 should be reworded, repetitions appear there, the purpose of the current manuscript is not clearly defined!
- Please indicate which statistical programs were used to evaluate the data and exactly what type of analyzes were used.
- In the discussion the Authors claimed that: “However, the findings also show that despite decreases observed globally, no progress has been made in reducing the age-standardized prevalence of HED among the total adult population and current drinkers in the WHO European Region”. However, the Authors do not provide accurate information on this subject. In my opinion, providing data on the distribution of drinking and changes in individual age groups could significantly enrich the work and provide a good basis for using the data e.g. to develop preventive programs.
- In the discussion, the Authors could also refer to the effectiveness of programs introduced by various countries. whether the statement that "the majority of the countries (33 out of 51) in the WHO European Region that were included in this study did not meet the goal of an at least 10% relative reduction in the harmful use of alcohol in either of the two indicators investigated" means that the governments of these countries do not use sufficient methods/ prophylactic programs to reduce the amount of consumed alcohol, or despite using various programs, the population is not responding.
- Please correct a few punctuation errors and errors in references.
- The answer to the question contained in the title should be included in the summary. For example, are countries doing something extra where consumption is not reduced?
Author Response
Meeting the global NCD target of at least 10% relative reduction in the harmful use of alcohol: is the WHO European Region on track?
Reviewer comments
Reviewer 1
Alcohol is a psychoactive substance with dependence-producing properties that has been widely used in many cultures for centuries. The harmful use of alcohol causes a large disease, social and economic burden in societies. The health, safety and socioeconomic problems attributable to alcohol can be reduced and requires actions on the levels, patterns and contexts of alcohol consumption and the wider social determinants of health. Countries have a responsibility for formulating, implementing, monitoring and evaluating public policies to reduce the harmful use of alcohol. Therefore, presented manuscript concerns a very current and important problem. The entire manuscript is written in accordance with the requirements of the journal. However, there are several weaknesses and the improvement of them can significantly increase value of manuscript.
Comment: In introduction the Authors claimed that “adult per capita consumption in liters of pure alcohol consumed per adult 15 years of age and older” but 15-year-old is not an adult. This sentence should be corrected.
Reply: 15+ is a definition of adulthood that is commonly used, not least by organizations such as the WHO and the World Bank. We, therefore, maintained the current definition in the manuscript.
Comment: If the authors use an abbreviation once, they should use it everywhere.
Reply: We revised all abbreviations to ensure consistent use throughout the manuscript. Note that abbreviations are re-introduced in tables, figures and the abstract as these should be able to stand alone. Furthermore, we do not use abbreviations in headings. Finally, we had to make a distinction between the WHO European Region (which has to be written out like this following WHO guidelines) and the subgroups defined in the manuscript for which we use the shorthand EUR.
Comment: Please explain the purpose of description from line 60 to line 70. In my opinion it should be deleted.
Reply: The paragraph summarizes the global frameworks and work programs that are currently accounting for alcohol consumption and its impact on noncommunicable disease burden. As the Global Action Plan for the Prevention and Control of NCDs sets out a clearly defined target to evaluate progress in the reduction of the harmful use of alcohol, we used this target for our
analyses. However, it is important to know that the other international frameworks such as the Sustainable Development Goals or the 13th General Programme of Work of the WHO have similar goals with regard to alcohol consumption. The paragraph was reword to improve readability.
Comment: Introduction is too short and general, it contains too few messages. It should be extended and include more information’s.
Reply: We added more information on the connection between alcohol use and NCDs and the NCD burden attributable to alcohol use in the Region.
Comment: Lines 60 to 80 should be reworded, repetitions appear there, the purpose of the current manuscript is not clearly defined!
Reply: The section was reworded to explain the three main frameworks and their targets more clearly. The purpose of the manuscript is stated in the last paragraph of the introduction section: “This study aims to explore the progress in achieving the target outlined in the Global Action Plan for the Prevention and Control of NCDs of reducing the harmful use of alcohol by 10% by 2025 concerning two targets: a) total adult per capita consumption APC and b) the age-standardized prevalence of HED by using estimates for the period 2010-2017 at regional and the country level.”
Comment: Please indicate which statistical programs were used to evaluate the data and exactly what type of analyzes were used.
Reply: We added the information that R statistical software was used (R version 3.6.1) to the methods section.
Comment: In the discussion the Authors claimed that: “However, the findings also show that despite decreases observed globally, no progress has been made in reducing the age-standardized prevalence of HED among the total adult population and current drinkers in the WHO European Region”. However, the Authors do not provide accurate information on this subject. In my opinion, providing data on the distribution of drinking and changes in individual age groups could significantly enrich the work and provide a good basis for using the data e.g. to develop preventive programs.
Reply: The statement cited above is supported by the finding that all confidence intervals for the changes in the age-standardized prevalence of HED included 0% change as described in the results section (“Overall, the findings show that little to no progress has been made regarding this indicator with all CIs including 0% change”). Analyzing developments in specific age groups was outside the scope of this study which focused on the overall progress towards international alcohol reduction targets.
Comment: In the discussion, the Authors could also refer to the effectiveness of programs introduced by various countries. whether the statement that "the majority of the countries (33 out of 51) in the WHO European Region that were included in this study did not meet the goal of an at least 10% relative reduction in the harmful use of alcohol in either of the two indicators investigated" means that the governments of these countries do not use sufficient methods/ prophylactic programs to reduce the amount of consumed alcohol, or despite using various programs, the population is not responding.
Reply: The discussion section lays out the most effective alcohol policies and several examples of policies that were implemented in specific countries that are known to be effective in reducing alcohol consumption and that likely contributed to the observed reductions. However, causal analyses explaining changes in alcohol consumption in specific countries require much more detailed analyses that were not feasible within this project.
Comment: Please correct a few punctuation errors and errors in references.
Reply: Done.
Comment: The answer to the question contained in the title should be included in the summary. For example, are countries doing something extra where consumption is not reduced?
Reply: The question contained in the title is now answered in the abstract (“Between 2010 and 2017 the reduction target for APC was met with a decline by -12.4% (95% confidence interval (CI) -17.2, -7.0%) in the region.”)
Reviewer 2 Report
The manuscript is clearly written, easily understandable, and concise. Only some punctuation mistakes are left.
Author Response
Thank you. We corrected the remaining mistakes in punctuation and spelling.
Reviewer 3 Report
This is an interesting and useful paper. A few points to consider:
p. 2 around line 69: You are given this 'goal' a pass, even though it includes some apparently nonsensical language - "as appropriate, within the national context." Is it a goal or not? What does context have to do with it?
It's an important point given that the first two indicators don't seem to consider context (are you arguing that they do?). I'm not sure C does either.
p. 2 around line 86 - are tourists associated with harmful alcohol use, more/different than residents?
p. 3 with regard to country groups, aside from their being WHO-level country groups and the like, is their use in this way for this study appropriate? Are there within-group similarities or is there reason to challenge the assumption for this particular study?
p. 3 if the goal was met, was that due to the goal or some other reason? How would one know?
p. 4 to 5 - if certain nations are responsible for the reduction, why look at it from a group perspective?
This goes back to the question of context and the 'why' of increases and decreases in consumption. Also the relevance of a goal, outside calling some public attention to a problem (and it's questionable how much attention is drawn as a result).
p. 9: if goal is met already then was the goal established well in the first place? This does not reduce my uncertainty about how the problem is being understood, defined, and addressed - if anything, my uncertainty is increased. I'm not sure how a group-based approach makes the most sense and I'm largely unconvinced. But perhaps that's more a policy question. Perhaps some of this can be addressed in the limitations, because the regional look is a useful addition to the literature, if problematic.
Author Response
Meeting the global NCD target of at least 10% relative reduction in the harmful use of alcohol: is the WHO European Region on track?
Reviewer comments
Reviewer 3
This is an interesting and useful paper. A few points to consider:
Comment: p. 2 around line 69: You are given this 'goal' a pass, even though it includes some apparently nonsensical language - "as appropriate, within the national context." Is it a goal or not? What does context have to do with it?
Reply: The Global Action Plan provides the WHO Member States with a road map and menu of policy options to collectively reduce premature mortality from NCDs by 2025 by 25%. In such international frameworks, the sovereignty of the member states in determining which policies to implement is affirmed through language such as “as appropriate, within the national context”. This expression is used several times in the Global Action Plan in different contexts. However, what we evaluated is the target itself and not the degree to which a country perceives the appropriateness of the target.
Comment: It's an important point given that the first two indicators don't seem to consider context (are you arguing that they do?). I'm not sure C does either.
Reply: See comment above.
Comment: p. 2 around line 86 - are tourists associated with harmful alcohol use, more/different than residents?
Reply: Tourists are not necessarily associated with more or less harmful alcohol use. This section merely describes how the total alcohol per capita consumption is defined and calculated. Tourist consumption is calculated for example to avoid overestimating consumption in countries with high levels of tourism but lower consumption in the general population of the country.
Comment: p. 3 with regard to country groups, aside from their being WHO-level country groups and the like, is their use in this way for this study appropriate? Are there within-group similarities or is there reason to challenge the assumption for this particular study?
Reply: The analysis of country groups can help draw a picture of broad differences within the WHO European Region without looking at single countries, such as the differences between changes in the EU-28 grouping and the Eastern WHO EUR grouping. The country groups are
based on geographic proximity, historic similarities in drinking patterns, and economic aspects. The groups were used in this study to increase comparability with previous studies and reports that have described the groupings in more detail [1,2].
Comment: p. 3 if the goal was met, was that due to the goal or some other reason? How would one know?
Reply: The objective of the study was not to measure the effect of setting the NCD targets on alcohol consumption but to provide a detailed progress report towards reaching the target of reducing the harmful use of alcohol by 10%. While the discussion section focused on the alcohol control policies that may have contributed to the observed changes, we now added a paragraph to the discussion section to acknowledge the potential role of other factors.
Comment: p. 4 to 5 - if certain nations are responsible for the reduction, why look at it from a group perspective?
Reply: This study looked at three levels of aggregation that provide different perspectives. The regional level allows for a comparison on the global level with other WHO regions, the country groups allow for an understanding of broad trends within the region and the country level allows for the most disaggregated, detailed picture (which on the other side may make it difficult to extract the broader trends).
Comment: This goes back to the question of context and the 'why' of increases and decreases in consumption. Also the relevance of a goal, outside calling some public attention to a problem (and it's questionable how much attention is drawn as a result).
Reply: See above. A paragraph was added to the discussion section to acknowledge other factors that may explain the observed changes.
Comment: p. 9: if goal is met already then was the goal established well in the first place? This does not reduce my uncertainty about how the problem is being understood, defined, and addressed - if anything, my uncertainty is increased. I'm not sure how a group-based approach makes the most sense and I'm largely unconvinced. But perhaps that's more a policy question. Perhaps some of this can be addressed in the limitations, because the regional look is a useful addition to the literature, if problematic.
Reply: As detailed in the manuscript, the target was met (seven years after it was established) by some countries and country groups, but not by all. The progress towards the target is not homogeneous and in the discussion section, we highlight several factors that likely contributed to favorable and swift development in some countries and country groups.
References
1. Shield, K.; Rylett, M.; Rehm, J., Public health successes and missed opportunities - Trends in alcohol consumption and attributable mortality in the WHO European Region, 1990–2014. World Health Organization: Copenhagen, Denmark, 2016.
2. Rehm, J.; Manthey, J.; Shield, K. D.; Ferreira-Borges, C., Trends in substance use and in the attributable burden of disease and mortality in the WHO European Region, 2010-2016. European Journal of Public Health 2019, 29, 723-8.
Reviewer 4 Report
The submitted manuscript describes some very interesting facts regarding the progress of alcohol consumption and the rate of episodes related to it. Overll the authors have made significant work trying to interpret their findings. However, although they establish the measures that can be taken from local governments and international organizations, such as EU, they do not mention other factors that may drive these results. For example they report that the reduction rates are higher in eastern and south-eastern couintries. Besides the governemental policies could be other factors that have aid in this reduction? Could it be that there is increased public information/education in these countries nowadays? Could it be the financial problems that maybe some of these countries are facing? Maybe this is not the scope of the manucript however, the implementation of these factors could assist in the adoption of similar techniques to countries that used to acquire these kind of policies but maybe nowadays they have been relaxed. Additionally, some minor comments below: Line 101-102: the authors mention that estimates of APC for the presented period are a combination of both collected and forecasted data. How did the authors combine these data? Did they use the mean estimate or they applied weights to take into account that the forecasted data might have more uncertainty than the empirical data (collected data)? Table 2 and Table 3: The 95% CI are supposed to be symmetric. However most of them is asymmetric. Being symmetric for a CI is particularly useful in case other researchers cite this manuscript to use the data.Author Response
Meeting the global NCD target of at least 10% relative reduction in the harmful use of alcohol: is the WHO European Region on track?
Reviewer comments
Reviewer 4The submitted manuscript describes some very interesting facts regarding the progress of alcohol consumption and the rate of episodes related to it. Overall, the authors have made significant work trying to interpret their findings.
Reply: Thank you.
Comment: However, although they establish the measures that can be taken from local governments and international organizations, such as EU, they do not mention other factors that may drive these results. For example, they report that the reduction rates are higher in eastern and south-eastern countries. Besides the governmental policies could be other factors that have aid in this reduction? Could it be that there is increased public information/education in these countries nowadays? Could it be the financial problems that maybe some of these countries are facing? Maybe this is not the scope of the manuscript however, the implementation of these factors could assist in the adoption of similar techniques to countries that used to acquire these kind of policies but maybe nowadays they have been relaxed.
Reply: We have added a paragraph to the discussion section to acknowledge that other factors such as economic growth/decline, unemployment rates, and social norms may have contributed to the observed changes.
Comment: Line 101-102: the authors mention that estimates of APC for the presented period are a combination of both collected and forecasted data. How did the authors combine these data? Did they use the mean estimate or they applied weights to take into account that the forecasted data might have more uncertainty than the empirical data (collected data)?
Reply: Where available, observed data were used. However, for the year 2017 only forecasted data were available. We added a sentence to the methods section to explain how uncertainty in the lower level estimates (the input data) was accounted for in the confidence intervals. Uncertainty in the lower level estimates is described on page 3, line 117 and following.
Comment: Table 2 and Table 3: The 95% CI are supposed to be symmetric. However, most of them is asymmetric. Being symmetric for a CI is particularly useful in case other researchers cite this manuscript to use the data.
Reply: 95% CIs do not need to be necessarily symmetric as they depend on the distribution of the variable. For normally distributed variables (most of the results reported), the 95% CI is indeed symmetric. Only for forecasted APC data (second column in Table 2), the interval towards the upper bound is larger as the forecasts were performed on logarithmized values (for illustration, see Figure 3 in the source publication [1]).
References
1. Manthey, J.; Shield, K.; Rylett, M. A.; Hasan, O. S. M.; Probst, C.; Rehm, J., Global alcohol exposure between 1990 and 2017 and forecasts until 2030: a modelling study. Lancet 2019, 393, 2493-2502.